# COVID-19 and Pulmonary Embolism Outcomes among Hospitalized Patients in the United States: A Propensity-Matched Analysis of National Inpatient Sample

**DOI:** 10.3390/vaccines10122104

**Published:** 2022-12-08

**Authors:** Adeel Nasrullah, Karthik Gangu, Nichole B. Shumway, Harmon R. Cannon, Ishan Garg, Hina Shuja, Aniesh Bobba, Prabal Chourasia, Abu Baker Sheikh, Rahul Shekhar

**Affiliations:** 1Division of Pulmonology and Critical Care, Allegheny Health Network, Pittsburg, PA 15212, USA; 2Department of Internal Medicine, University of Kansas Medical Center, Kansas City, KS 66160, USA; 3Department of Internal Medicine, University of New Mexico Health Sciences Center, Albuquerque, NM 87106, USA; 4Department of Medicine, Karachi Medical and Dental College, Karachi 74700, Pakistan; 5Department of Medicine, John H Stronger Hospital, Cook County, Chicago, IL 60612, USA; 6Department of Medicine, Mary Washington Hospital, Fredericksburg, VA 22401, USA

**Keywords:** COVID-19, pulmonary embolism, venous thromboembolism, National Inpatient Sample

## Abstract

Venous thromboembolism, in particular, pulmonary embolism (PE), is a significant contributor to the morbidity and mortality associated with COVID-19. In this study, we utilized the National Inpatient Sample (NIS) database 2020 to evaluate and compare clinical outcomes in patients with COVID-19 with and without PE. Our sample includes 1,659,040 patients hospitalized with COVID-19 pneumonia between January 2020 and December 2020. We performed propensity-matched analysis for patient characteristics and in-hospital outcomes, including the patient’s age, race, sex, insurance status, median income, length of stay, mortality, hospitalization cost, comorbidities, mechanical ventilation, and vasopressor support. Patients with COVID-19 with PE had a higher need for mechanical ventilation (25.7% vs. 15.6%, adjusted odds ratio 1.4, 95% CI 1.4–1.5, *p* < 0.001), the vasopressor requirement (5.4% vs. 2.6%, adjusted OR 1.6, 95% CI 1.4–1.8, *p* < 0.001), longer hospital stays (10.8 vs. 7.9 days, *p* < 0.001), and overall higher in-hospital mortality (19.1 vs. 13.9%, adjusted OR of 1.3, 95% CI 1.1–1.5, *p* < 0.001). This study highlights the need for more aggressive management of PE in COVID-19-positive patients with the aim to improve early diagnosis and treatment to reduce morbidity, mortality, and healthcare costs seen in the synchronous COVID-19 and PE-positive patients.

## 1. Introduction

The coronavirus disease-19 (COVID-19), caused by SARS-CoV-2 virus is a multiorgan disease with a wide range of presentations from mild flu-like symptoms to severe acute respiratory distress syndrome (ARDS) and thrombotic events involving arterial and venous circulation. As of October 2022, more than one million patients have died of COVID-19 [1]. Venous thromboembolism, in particular pulmonary embolism (PE), is a significant contributor to morbidity and mortality associated with COVID-19. In comparison to influenza, COVID-19 leads to a more pronounced hypercoagulable state secondary to endotheliopathy, and a hyperinflammatory state which results in the formation of venous and arterial thrombosis [2]. Profound inflammatory state (cytokine storm) has been described in COVID-19 which results in elevated C-reactive protein, interleukin-6, and serum ferritin. Systemic inflammation and direct endotheliopathy results in activation of platelets and endothelium leading to a procoagulant state and formation of thrombi [3]. PE incidence is strongly associated with the severity of COVID-19, with a higher incidence in critically ill patients. Based on the prior literature, patients with COVID-19 have a seven-fold higher risk of PE than non-COVID patients [3]. At the onset of the pandemic, due to limited understanding, there was a lack of a systematic approach to the diagnosis and management of PE in COVID-19. Hence, the reported incidence varied notably. A Cochrane review reported the incidence of PE in hospitalized patients as 4.3%, while other studies reported it ranging from 2.2 to 8.3% [4]. Despite venous thromboprophylaxis, patients with COVID-19 tend to develop PE as compared to non-COVID-related PE. Hence, various studies have described patient characteristics associated with the development of PE in patients with COVID-19 such as male sex, obesity, intensive care unit admission, and elevated D-dimers [5]. There is no large-scale study comparing outcomes in Patients with COVID-19 with and without PE.

In our study, we utilized the National Inpatient Sample (NIS) database to evaluate and compare clinical outcomes in patients with COVID-19 with and without PE. Moreover, we performed a propensity-matched analysis to assess the effect of PE on various clinical outcomes.

## 2. Materials and Methods

This was a retrospective study conducted using the N.I.S. data from 2020. Established by the agency for healthcare research and quality (AHRQ), the N.I.S. is the largest all-payer healthcare database in the United States. The international classification of diseases 10th—clinical modification (ICD-10-CM) codes were used to retrieve patient samples with comorbid conditions, and ICD-10 procedure codes were used to identify inpatient procedures. All patients who were 18 years of age and older and admitted to the hospital with COVID-19 infection were included in this study.

The N.I.S. database contains data regarding in-hospital outcomes, procedures, and other discharge-related information. Variables were divided into patient-related, hospital-related, and indicators of illness severity as below:Patient: age, race (white, black, Hispanic, Native American, Asian, other), sex, insurance status (Medicare, Medicaid, private insurance, self-payment, no charge), median income based on patient’s zip code, and disposition.Hospital: location, teaching status, bed size, and region.Illness severity: length of stay (L.O.S.), mortality, hospitalization cost, comorbidities, mechanical ventilation, circulatory support, and vasopressor use.

The primary outcome was in-hospital mortality. Secondary outcomes included:Intubation and mechanical ventilation;Vasopressor use;Length of stay, the financial burden on healthcare, and resource utilization.

### Statistical Analysis

Descriptive statistics were used to summarize the continuous and categorical variables. Continuous variables were summarized as mean ± SD; categorical data as number and percentage. Univariate analyses for between-group comparisons used the Rao–Scott chi-square test for categorical variables (e.g., sex and risk factors) and weighted simple linear regression for continuous variables (e.g., age). One unmatched sample univariate regression was used to identify independent variables (*p* ≤ 0.2), which were utilized to build a multivariate regression model. As our control group (COVID-19 positive without PE) had a significantly higher sample than test group (COVID-19 positive with PE), we conducted a secondary analysis with propensity matching to confirm results obtained by a traditional multivariate analysis. Baseline demographics (age, race, sex, income status, insurance status) were matched using 1:1 nearest neighbor propensity score with 0.05 caliper width. On matched cohort a secondary multivariate regression model was built as described above. All analysis was performed using Stata software version 17.0 (Stata Corporation, College Station, TX, USA). *p* values of less than 0.05 were considered statistically significant.

## 3. Results

### 3.1. Demographics and Baseline Comorbidities

A total of 1,659,040 patients with COVID-19 were hospitalized between 1 January and 31 December 2020, of which, 46,440 (2.8%) were diagnosed with a PE. The age-group in which COVID-19 infection with PE was most common was 50–69 years old (42.83% vs. 37.97%, *p* < 0.001). Caucasians and African Americans with COVID-19 appeared to have a higher risk of developing a PE compared to other races (53.31% vs. 50.84%, *p* < 0.001 and 23.23% vs. 18.93%, *p* < 0.001, respectively). Notably, Hispanics with COVID-19 had lower rates of PE (21.62% vs. 16.42%, *p* < 0.001). Income and insurance status appeared to be generally equal throughout all groups (Table 1).

There was not a significant difference between patients with COVID-19 with PE and without PE in regards to chronic pulmonary disease (21.62% vs. 22.01%, *p* = 0.38) and smoking status (25.87% vs. 25.59%, *p* = 0.56). Patients with COVID-19 who presented with a PE had higher rates of uncomplicated hypertension (40.81% vs. 38.05%, *p* < 0.001) and obesity (27.91% vs. 25.57%, *p* < 0.001). Interestingly, patients with COVID-19 without PE had higher rates of CAD (18.06% vs. 14.46%, *p* < 0.001), CHF (17.6% vs. 16.39%, *p* = 0.002), complicated hypertension (27.21% vs. 21.77%, *p* < 0.001), uncomplicated diabetes mellitus (14.77% vs. 13.7%, *p* = 0.004), complicated diabetes mellitus (26.46% vs. 22.63%, *p* < 0.001), and renal failure (21.22% vs. 14.7%, *p* < 0.001) when compared to the COVID-19 with PE cohort. (Table 1) (Figure 1).

Propensity matching was performed in regards to patient age, sex, race, income, and insurance status. After PSM, we had a total of 41,975 patients in each group (COVID-19 negative and COVID-19 positive). (Table 2).

### 3.2. In-Hospital Mortality

COVID-19 infection and PE had significantly higher rates of in-hospital mortality compared to patients with COVID-19 without PE (19.24% vs. 13.25%, adjusted OR 1.36 [95% CI 1.28–1.45, *p* < 0.001]). Mortality was higher in COVID-19 positive African Americans with PE compared to non-PE (18.59% vs. 16.8%, *p* = 0.05). There was no difference in in-hospital mortality in regards to PE vs. non-PE in Hispanics (21% vs. 19.53%, *p* = 0.19) or Native Americans (1.46% vs. 1.21%, *p* = 0.46). COVID-19 positive Caucasians without PE had higher mortality compared to PE group (51.64% vs. 47.82%, *p* = 0.004). Patients with COVID-19 aged 30–49 (8.17% vs. 4.8%, *p* < 0.001) and 50–69 (39.47% vs. 30.15%, *p* < 0.001) with PE also had higher rates of in-hospital mortality compared to the non-PE cohort. (Table 3 and Table 4) (Figure 2 and Figure 3).

PSM results for COVID-19-positive patients with and without PE were similar to the multivariate analysis in regards to in-hospital mortality. Patients with PE had significantly higher rates of in-hospital mortality (19.09% vs. 13.94%, adjusted OR 1.29 [95% CI 1.13–1.47, *p* < 0.001]). (Table 5).

### 3.3. In-Hospital Complications

COVID-19-positive patients with PE required higher rates of mechanical ventilation (25.68% vs. 15.62%, adjusted OR 1.44 [95% CI 1.36–1.52, *p* < 0.001]), higher vasopressor use (5.4% vs. 2.57%, adjusted OR 1.62 [95% CI 1.45–1.81, *p* < 0.001]), higher rates of cardiogenic shock (1.78% vs. 0.59%, adjusted OR 2.15 [95% CI 1.80–2.57, *p* < 0.001]), higher use of mechanical circulatory support (LVAD, pVAD, ECMO) (0.83% vs. 0.26%, adjusted OR 2.03 [95% CI 1.53–2.69, *p* < 0.001]), and sudden cardiac death (5% vs. 2.67%, adjusted OR 1.53 [95% CI 1.38–1.70])(Table 2).

Again, PSM results for COVID-19-positive patients with and without PE were similar to the unmatched numbers in regards to use of mechanical ventilation (25.18% vs. 16.28%, adjusted OR 1.55 [95% CI 1.36–1.76, *p* < 0.001]), vasopressor use (5.35% vs. 3.78%, adjusted OR 1.79 [95% CI 1.41–228, *p* < 0.001]), cardiogenic shock (1.8% vs. 0.6%, adjusted OR 3.1 [95% CI 2.02–4.75, *p* < 0.001]), mechanical circulatory support (LVAD, pVAD, ECMO) (0.82% vs. 0.3%, adjusted OR 3.1 [95% CI 1.67–5.76, *p* < 0.001]), and sudden cardiac death (5% vs. 2.4%, adjusted OR 1.44 [95% CI 1.11–1.88])(Table 4).

### 3.4. In-Hospital Quality Measures and Disposition

COVID-19 positive patients with PE had increased mean length of stay (10.8 days vs. 7.9 days, adjusted length of stay of 1.9 days higher, *p* < 0.001) compared to patients with COVID-19 without PE. They also had a higher mean total hospitalization cost (143,996 USD vs. 90,337 USD, adjusted total cost 38,907 USD higher, *p* < 0.001). Of those patients who survived, fewer patients in the PE group were able to discharge home (56.15% vs. 61.19%, *p* < 0.001) and required skilled nursing or long-term acute care compared to the non-PE group (23.73% vs. 22.15%, *p* < 0.001)(Table 2)(Figure 2 and Figure 3).

In regards to PSM results, findings were again similar to the unmatched data set with notable difference in disposition status. COVID-19 positive patients with PE were noted to have increased the mean length of stay (10.8 days vs. 8.4 days, adjusted length of stay 2.3 days higher, *p* < 0.001). They also had increased mean total hospitalization costs (143,705 USD vs. 78,407 USD, adjusted total cost 52,024 USD higher, *p* < 0.001). Interestingly, in regards to disposition status, the PSM results showed that COVID-19-positive patients with PE were more likely to discharge home (55.13% vs. 52.74%, *p* < 0.001) and less likely to go to skilled nursing or long-term acute care (24.39% vs. 25.95%, *p* < 0.001) when compared to the COVID-19 positive non-PE cohort (Table 4).

## 4. Discussion

To our knowledge, this is the largest recent national cohort assessing outcomes of COVID-19 with pulmonary embolism (WPE) versus without PE. It includes 1,659,040 patients hospitalized with COVID-19 pneumonia between January 2020 till December 2020. The most salient findings of our study include: (1) COVID-19 WPE was noted to have significantly higher in-hospital mortality as compared to COVID-19 without PE. (2) Patients with COVID-19 WPE were noted to have a higher incidence of sudden cardiac arrest, cardiogenic shock, and need for mechanical circulatory support as compared to patients with COVID-19 without PE. (3) Patients with COVID-19 WPE required considerably more vasopressor and mechanical ventilation support as compared to patients with COVID-19 without PE. (4) Overall hospital length of stay and cost associated with hospitalization was substantially higher in patients with COVID-19 WPE in contrast to patients with COVID-19 without PE. (5) Male sex and old age predicted a higher incidence of mortality in patients with COVID WPE and without PE, however, there was no association of mortality with race, household income, or insurance status.

Currently, the reported incidence of PE in COVID-19 pneumonia is highly variable due to the lack of a systematic approach to diagnosis and heterogeneity in reporting of the available literature. Based on the currently available literature, the incidence of PE in COVID-19 ranges from 2.6–47% with a higher incidence in patients who require intensive care [3,6,7,8]. Our study reports an incidence of PE as 2.8% in COVID-19 disease which is lower than the prior reported literature. It could be explained by the inclusion of all hospitalized patients with COVID-19 regardless of the disease severity. The higher incidence of PE in patients with severe COVID-19 who require mechanical ventilation, vasopressor, or circulatory support may be explained by the patient and disease-related factors such as immobility, age, comorbidities, central venous catheters, sepsis, and hyperinflammatory state.

COVID-19 with PE is associated with higher odds of in-hospital mortality as seen in prior studies [9]. We report in-hospital mortality of 19.09% in our study which is significantly less than prior studies, as reported up to 43.2–50% [10,11]. This difference could be explained by multiple factors i.e., sample size, selection bias, the paradigm shift in the management of COVID-19 pneumonia with the recommendation of steroid use as per the RECOVERY trial published in June 2020 [12], and different virus strains with varying disease severity. A large retrospective cohort from Spain has reported mortality of up to 16% in COVID WPE which is comparable to our findings [13]. Similarly, a New York Health System reported patients with COVID-19 WPE had a mortality rate of 20%, which is parallel to our study [14]. Cossio et al. reported that in comparison to non-COVID PE, Patients with COVID-19 with PE were less likely to have classic risk factors and symptoms of venous thromboembolism, however no mortality difference was noted at 1 year [15]. Similarly, Miro et al. replicated the same results as Cossio et al., however in-hospital mortality was 16% in COVID-19-related PE vs. non-COVID PE. The difference in mortality in these true groups could be related to the severity of COVID-19 and secondary complications of ARDS, hence the true role of COVID-19 determining PE outcomes remains to be elucidated by further studies. Notably, COVID-19 vaccines were approved in December 2020, which has significantly decreased the morbidity and mortality associated with COVID-19 disease, hence the effect of vaccine administration may be underestimated in our study due to the study period [16]. Law et al. reported that unvaccinated patients had a 2.75-fold increased risk of COVID-19-associated PE. The incidence of PE was notably high with the ancestral strain of COVID-19 (15%) as compared to Delta and Omicron variants (10.6% and 9.3% respectively). However, the incidence of PE was still lower in vaccinated patients during Delta and Omicron variants periods [17]. During our study period, the use of anticoagulation was heterogeneous, depending on institutions and physician discretion, hence a conclusive effect of anticoagulation on mortality cannot be drawn. As more robust evidence was available, the National Institute of Health recommended the utilization of therapeutic anticoagulation in non-critically ill COVID-19 disease (suggested benefit in favor of organ support-free days and mortality) and prophylactic anticoagulation in severe, critically ill COVID-19 disease due to a high risk of bleeding [18,19].

Cardiogenic shock can result from a hyperinflammatory state, high-risk pulmonary embolism, acute coronary syndrome, stress cardiomyopathy, or viral myocarditis, which have been reported in COVID-19 disease [20,21]. Our study found that patients with COVID-19 and PE are three times more likely to go into cardiogenic shock as compares to patients only with COVID-19. Prior studies have reported the incidence of cardiogenic shock from 9.1–37%, which is likely overestimated due to limited sample sizes and selection bias [11,22,23,24]. Similarly, Nakamura et al. reported cardiogenic shock incidence as 36% in pulmonary embolism patients without COVID-19 [25]. Such differences could be explained by higher comorbidity index, and central thrombosis seen in non-COVID-19 PE patients. We report the need for mechanical ventilation (MV) to 25.18% in patients with COVID-19 WPE. Prior reported studies had equivocal statistical and clinical significance in regards to the need for MV in patients with COVID-19 without PE or with PE had a reported need for MV in up to 20–67% of cases [14,26]. Our study indicates a statistically significant higher need for mechanical ventilation in patients with WPE, which could be explained by the degree of ventilation/perfusion mismatch, higher incidence of PE in critically ill patients, and higher power of the study.

Limited case series have reported sudden cardiac arrest in association with COVID-19 pneumonia with underlying suspected PE, which confers high mortality [27]. Our study reports an incidence of cardiac arrest of up to 5% in COVID-19 WPE versus 2.4% without PE. A prompt systemic approach to diagnosis and interventional therapy including thrombolytics and or catheter-based therapies may be necessary to mitigate mortality associated with PE-related obstructive shock. Wang et al. reported the incidence of shock in COVID-19 as 8.7%, which is parallel to our study (9.31%) [28]. Similarly, and notably, COVID-19 WPE required more vasopressor support (5.3%) as compared to without PE (3.7%), which could be explained by multifactorial shock i.e., distributive, obstructive, and/or cardiogenic.

Refractory hypoxic respiratory failure and acute cor pulmonale may require an institution of mechanical circulatory support (MCS). John et al. and Mariani et al. have comprehensively reviewed the utilization of MCS in COVID-19 and reported venovenous extracorporeal membrane oxygenation in conjunction with an impella being the most common device used with favorable outcomes [24,29]. However, no observational study has explicitly reviewed MCS need in COVID-19 WPE. Our study indicated a higher utilization of MCS in patients with WPE, likely due to acute cor pulmonale.

Our study reports that COVID-19 WPE is associated with a longer hospital stay of 10.8 vs. 8.4 days as compared to without PE. This is consistent with prior reported evidence which suggests hospital length of stay of up to 7 days [22]. Similarly, our study indicates the mean hospitalization cost for COVID-19 WPE is approximately twice of patients without PE, which is likely in the setting of longer hospital stay with increased utilization of treatment modalities.

Our study has several limitations. The first is inherent limitations due to the retrospective nature of the study. The ICD-10 diagnosis code for COVID-19 was not developed until late 2020, which allows for the possibility that certain COVID-19 diagnoses were not captured in our dataset. Likewise, COVID-19 testing remained in short supply and the NIS database does not provide lab test results to confirm COVID-19 status. Secondly, the data was retrieved using ICD-10 codes and therefore is prone for coding errors (see Appendix A). Our statistical analysis accounted for numerous demographic and clinical factors, including medical comorbidity, via multivariable logistic regression—however unaccounted variables may underlie our findings. This dataset did not include biochemical, certain clinical (vital signs), and radiographic information on PE and COVID-19 infection.

## 5. Conclusions

In our study, patients with COVID-19 with pulmonary embolism had a higher incidence of in-hospital mortality, sudden cardiac arrest, cardiogenic shock, need for vasopressors, mechanical ventilation, and mechanical circulatory support as compared to patients with COVID-19 without PE. Moreover, patients with COVID-19 with PE had a longer hospital stay with a significantly high financial burden associated with it. Hence, our study suggests prompt recognition and management of pulmonary embolism is essential to decrease morbidity, mortality, and health care costs associated with it.

## Figures and Tables

**Figure 1 vaccines-10-02104-f001:**
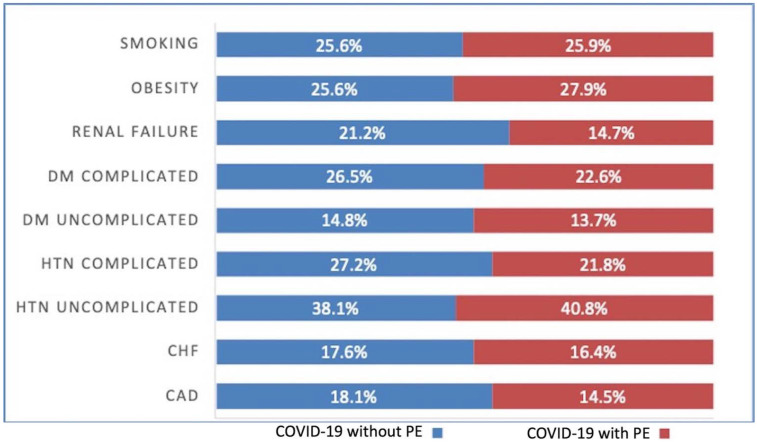
COVID-19 and Pulmonary embolism (PE) patient comorbidities. DM = Diabetes Mellitus, HTN = Hypertension, CHF = Congestive heart failure, CAD = Coronary artery disease, +ve = Positive, −ve = Negative.

**Figure 2 vaccines-10-02104-f002:**
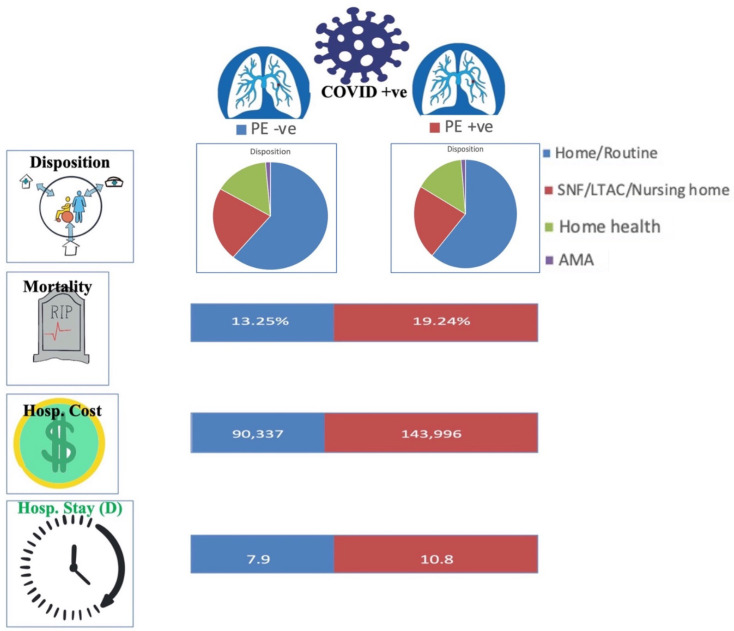
COVID-19 and Pulmonary embolism (PE) patient outcomes. SNF = skilled nursing facility, ALTAC = A long-term acute care, AMA = against medical advice, Hosp. = Hospital, D = days. +ve = Positive, −ve = Negative.

**Figure 3 vaccines-10-02104-f003:**
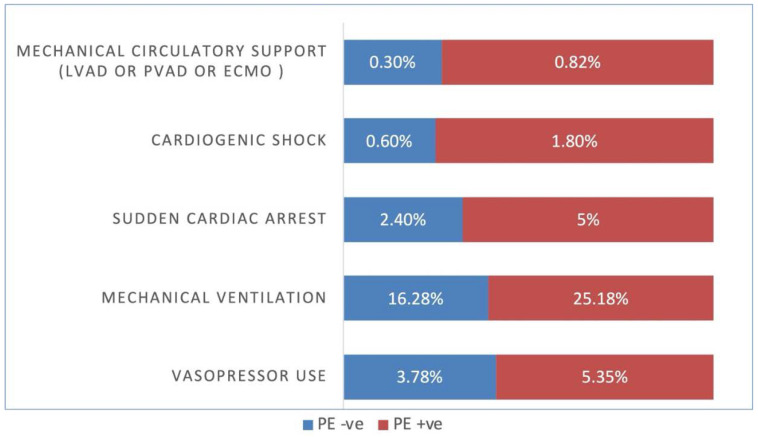
COVID-19 and Pulmonary embolism (PE) patient complications. LVAD = left ventricular assist device, PVAD = percutaneous ventricular assist device, ECMO = Extracorporeal membrane oxygenation, +ve = Positive, −ve = Negative.

**Table 1 vaccines-10-02104-t001:** COVID-19 and pulmonary embolism patient-level characteristics. SD = Standard deviation, CAD = Coronary Artery disease, CHF = Congestive heart failure, HTN = Hypertension, DM = Diabetes Mellitus.

Characteristics	COVID-19 without Pulmonary Embolism *n* (%)	COVID-19 with Pulmonary Embolism *n* (%)	*p* Value
*n* = 1,659,040	1,612,600 (97.2%)	46,440 (2.8%)	
**SEX (Female)**	789,368 (48.95%)	19,064 (41.05%)	<0.001
**Mean age years (SD)**			<0.001
Male	63.4 (16.3)	62.92 (14.9)	
Female	63.0 (18.9)	65.68 (15.9)	
**AGE GROUPS (%)**			<0.001
≥18–29	80,953 (5.02%)	1059 (2.28%)	
30–49	271,562 (16.84%)	7156 (15.41%)	
50–69	597,791 (37.07%)	19,890 (42.83%)	
≥70	662,295 (41.07%)	18,335 (39.48%)	
**RACE (%)**			<0.001
Caucasians	819,846 (50.84%)	24,757 (53.31%)	
African American	305,265 (18.93%)	10,788 (23.23%)	
Hispanics	348,644 (21.62%)	7625 (16.42%)	
Asian or Pacific Islander	52,732 (3.27%)	1073 (2.31%)	
Native American	16,610 (1.03%)	418 (0.9%)	
Others	69,342 (4.3%)	1765 (3.8%)	
**MEDIAN HOUSEHOLD INCOME (%)**			<0.001
<49,999$	550,542 (34.14%)	15,423 (33.21%)	
50,000–64,999$	438,143 (27.17%)	13,073 (28.15%)	
65,000–85,999$	357,352 (22.16%)	10,286 (22.15%)	
>86,000$	266,563 (16.53%)	7658 (16.49%)	
**INSURANCE STATUS (%)**			0.003
Medicare	859,354 (53.29%)	23,847 (51.35%)	
Medicaid	245,599 (15.23%)	5930 (12.77%)	
Private	443,787 (27.52%)	14,875 (32.03%)	
Self-pay	63,859 (3.96%)	1788 (3.85%)	
**HOSPITAL DIVISION (%)**			<0.001
New England	60,956 (3.78%)	1904 (4.1%)	
Middle Atlantic	235,762 (14.62%)	6627 (14.27%)	
East North Central	248,824 (15.43%)	8972 (19.32%)	
West North Central	108,044.2 (6.7%)	3864 (8.32%)	
South Atlantic	323,488 (20.06%)	9362 (20.61%)	
East South Central	108,205 (6.71%)	3144 (6.77%)	
West South Central	232,376 (14.41%)	5224 (11.25%)	
Mountain	111,269 (6.9%)	3334 (7.18%)	
Pacific	183,675 (11.39%)	3799 (8.18%)	
**HOSPITAL BEDSIZE (%)**			<0.001
Small	393,474 (24.4%)	10,124 (21.8%)	
Medium	46,733 (28.98%)	13,570 (29.22%)	
Large	751,633 (46.61%)	22,746 (48.98%)	
**HOSPTAL TEACHING STATUS (%)**			<0.001
Rural	158,196 (9.81%)	4458 (9.6%)	
Urban non-teaching	301,556 (18.7%)	7950 (17.12%)	
Urban teaching	1,152,848 (71.49%)	34,031 (73.28%)	
**COMORBIDITIES (%)**			
CAD	291,236 (18.06%)	6715 (14.46%)	<0.001
CHF	283,818 (17.6%)	7611 (16.39%)	0.002
HTN uncomplicated	613,594(38.05%)	18,952 (40.81%)	<0.001
HTN complicated	438,788 (27.21%)	10,110 (21.77%)	<0.001
DM uncomplicated	238,181 (14.77%)	6362 (13.7%)	0.004
DM complicated	426,694 (26.46%)	10,509 (22.63%)	<0.001
Renal failure	342,194 (21.22%)	6827 (14.7%)	<0.001
Chronic pulmonary disease	354,933 (22.01%)	10,040 (21.62%)	0.38
Obesity	412,342 (25.57%)	12,961 (27.91%)	<0.001
Smoking	412,664 (25.59%)	12,014 (25.87%)	0.56

**Table 2 vaccines-10-02104-t002:** Patients with COVID-19 with and without pulmonary embolism propensity 1:1 matched patient-level characteristics.

Characteristics	COVID-19 without Pulmonary Embolism	COVID-19 with Pulmonary Embolism	*p* Value
PE (83,950)	*n* = 41,975	*n* = 41,975	
**SEX (Female)**	17,684 (42.13%)	17,684 (42.18%)	0.95
**Mean age years (SD)**	64.50 (13.99)	64.45 (14.03)	0.87
**AGE GROUPS (%)**			0.97
≥18–29	940 (2.24%)	978 (2.33%)	
30–49	6170 (14.70%)	6149 (14.65%)	
50–69	17,835 (42.49%)	17,856 (42.54%)	
≥70	17,029 (40.57%)	16,991 (40.48%)	
**RACE (%)**			0.99
Caucasians	22,641 (53.94%)	2267 (54.02%)	
African American	9684 (23.07%)	9696 (23.10%)	
Hispanics	6804 (16.21%)	6833 (16.28%)	
Asian or Pacific Islander	995 (2.37%)	949 (2.26%)	
Native American	332 (0.79%)	294 (0.70%)	
Others	1519 (3.62%)	1524 (3.63%)	
**MEDIAN HOUSEHOLD INCOME (%)**			1
<49,999$	13,810 (32.90%)	13,801 (32.88%)	
50,000–64,999$	11,690 (27.85%)	11,711 (27.90%)	
65,000–85,999$	9390 (22.37%)	9394 (22.38%)	
>86,000$	7085 (16.88%)	7069 (16.84%)	
**INSURANCE STATUS (%)**			0.99
Medicare	21,638 (51.55%)	21,621 (51.51%)	
Medicaid	5293 (12.61%)	5327 (12.69%)	
Private	13,436 (32.01%)	13,461 (32.07%)	
Self-pay	1603 (3.82%)	1570 (3.74%)	
**HOSPITAL DIVISION (%)**			<0.001
New England	1742 (4.15%)	24,329 (57.96%)	
Middle Atlantic	6271 (14.94%)	15,606 (37.18%)	
East North Central	8370 (19.94%)	1184 (2.82%)	
West North Central	3077 (7.33%)	218 (0.52%)	
South Atlantic	8525 (20.31%)	365 (0.87%)	
East South Central	2909 (6.93%)	504 (1.20%)	
West South Central	4596 (10.95%)	101 (0.24%)	
Mountain	3026 (7.21%)	109 (0.26%)	
Pacific	3463 (8.25%)	55 (0.13%)	
**HOSPITAL BEDSIZE (%)**			<0.001
Small	9134 (21.76%)	12,836 (30.58%)	
Medium	12,265 (29.22%)	13,969 (33.28%)	
Large	20,576 (49.02%)	15,170 (36.14%)	
**HOSPTAL TEACHING STATUS (%)**			<0.001
Rural	3874 (9.23%)	1943 (4.63%)	
Urban non-teaching	7312 (17.42%)	4814 (11.47%)	
Urban teaching	30,789 (73.35%)	35,217 (83.90%)	

**Table 3 vaccines-10-02104-t003:** In-hospital outcomes of Patients with COVID-19 with and without pulmonary embolism. SNF = skilled nursing facility, ALTAC = A long-term acute care, AMA = against medical advice.

Variable	COVID-19 with Pulmonary Embolism	COVID-19 without Pulmonary Embolism	*p* Value
Disposition			<0.001
Home/Routine	26,076 (56.15%)	986,750 (61.19%)	
SNF/LTAC/Nursing home	11,020 (23.73%)	357,191 (22.15%)	
Home health	9019 (19.42%)	246,566.54 (15.29%)	
AMA	325 (0.7%)	22,093 (1.37%)	
**Vasopressor use**	2508 (5.4%)	41,444 (2.57%)	
	Adjusted odds ratio ^1^ 1.62 (95% CI 1.45–1.81)		<0.001
**Mechanical ventilation**	11,926 (25.68%)	251,888 (15.62%)	
	Adjusted odds ratio ^1^ 1.44 (95% CI 1.36–1.52)		<0.001
**Sudden Cardiac Arrest**	2322 (5%)	43,056 (2.67%)	<0.001
	Adjusted odds ratio ^1^ 1.53 (95% CI 1.38–1.70)		
**Cardiogenic Shock (%)**	827 (1.78%)	9514 (0.59%)	
	Adjusted odds ratio ^1^ 2.15 (95% CI 1.80–2.57)		<0.001
**Mechanical Circulatory Support (%)** **(LVAD or pVAD or ECMO)**	385 (0.83%)	4193 (0.26%)	
	Adjusted odds ratio ^1^ 2.03 (95% CI 1.53–2.69)		<0.001
**In-hospital mortality (*n* = 222,490) (%)**	8935 (19.24%)	213,555 (13.25%)	
	Adjusted odds ratio ^1^ 1.36 (95% CI 1.28–1.45)		<0.001
**Mean total hospitalization charge ($)**	143,996$	90,337$	
	Adjusted total charge ^1^ 38,907$ higher		<0.001
**Mean length of stay (days)**	10.8 (10.5–11.1)	7.9 (7.8–8.0)	
	Adjusted length of stay ^1^ 1.9 day higher		<0.001

^1^ Adjusted for age, sex, race, income level, insurance status, discharge quarter, elixhauser co-morbidities, hospital location, teaching status and bed size.

**Table 4 vaccines-10-02104-t004:** Mortality breakdown Patients with COVID-19 with and without pulmonary embolism (unmatched sample).

Variable	COVID-19 with Pulmonary Embolism	COVID-19 without Pulmonary Embolism	*p* Value
Total died (222,490)	8935	213,555	
**SEX (%)**			<0.001
Male	5663 (63.38%)	124,417 (58.%)	
Female	3272 (36.62%)	89,138 (41.7%)	
**AGE GROUPS (%)**			
≥18–29	70 (0.78%)	1260 (0.59%)	0.30
30–49	730 (8.17%)	10,251 (4.8%)	<0.001
50–69	3527 (39.47%)	64,387 (30.1%)	<0.001
≥70	4608 (51.57%)	137,658 (64.%)	<0.001
**RACE (%)**			
Caucasians	4273 (47.82%)	110,280 (51.%)	0.004
African American	1661 (18.59%)	35,877 (16.8%)	0.05
Hispanics	1876 (21%)	41,707 (19.5%)	0.19
Asian or Pacific Islander	240 (2.69%)	7218 (3.38%)	0.13
Native American	130 (1.46%)	2584 (1.21%)	0.41
Others	385 (4.31%)	9418 (4.41%)	0.85
**HOSPTAL TEACHING STATUS (%)**			
Rural	566 (6.33%)	16,849 (7.89%)	0.03
Urban non-teaching	1316 (14.73%)	38,162 (17.8%)	0.002
Urban teaching	7054 (78.95%)	158,565 (74%)	<0.001

**Table 5 vaccines-10-02104-t005:** In-hospital outcomes of Patients with COVID-19 with and without pulmonary embolism for 1:1 PS matched the sample. SNF = skilled nursing facility, ALTAC = A long-term acute care, AMA = against medical advice, Hosp. = Hospital, D = days. +ve = Positive, −ve = Negative.

Variable	COVID-19 with Pulmonary Embolism	COVID-19 without Pulmonary Embolism	*p* Value
Disposition			<0.001
Home/Routine	23,141 (55.13%)	22,138 (52.74%)	
SNF/LTAC/Nursing home	10,238 (24.39%)	10,892 (25.95%)	
Home health	8303 (19.78%)	8370 (19.94%)	
AMA	298 (0.71%)	575 (1.37%)	
**Vasopressor use (%)**	2246 (5.35%)	1587 (3.78%)	
	Adjusted odds ratio ^1^ 1.79 (95% CI 1.41–2.28)	<0.001	
**Mechanical ventilation (%)**	10,569 (25.18%)	6834 (16.28%)	
	Adjusted odds ratio ^1^ 1.55 (95% CI 1.36–1.76)	<0.001	
**Sudden Cardiac Arrest (%)**	2099 (5%)	1007 (2.4%)	
	Adjusted odds ratio ^1^ 1.44 (95% CI 1.11–1.88)	0.006	
**Cardiogenic Shock (%)**	756 (1.8%)	252 (0.6%)	
	Adjusted odds ratio ^1^ 3.1 (95% CI 2.02–4.75)	<0.001	
**Mechanical Circulatory Support (%)** **(LVAD or pVAD or ECMO)**	344 (0.82%)	126 (0.3%)	
	Adjusted odds ratio ^1^ 3.1 (95% CI 1.67–5.76)	<0.001	
**In-hospital mortality (%)** **(*n* = 13,865)**	8014 (19.09%)	5851 (13.94%)	
	Adjusted odds ratio ^1^ 1.29 (95% CI 1.13–1.47)	<0.001	
**Mean total hospitalization charge ($)**	143,705$	78,407$	
	Adjusted total charge ^1^ 52,024$ higher	<0.001	
**Mean length of stay (days)**	10.8	8.4	
	Adjusted length of stay ^1^ 2.3 day higher	<0.001	

^1^ Adjusted for discharge quarter, elixhauser co-morbidities, hospital location, teaching status, and bed size.

## Data Availability

Not applicable.

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
