# Peer review of "COVID-19 and Pulmonary Embolism Outcomes among Hospitalized Patients in the United States: A Propensity-Matched Analysis of National Inpatient Sample"

_vaccines, 2022, doi:10.3390/vaccines10122104_

Round 1

Reviewer 1 Report

I think that it is an interesting paper without particular areas of flags. Conclusions are coherent with methods and bibliography is appropriate. No Introduction is appropriate.

Author Response

We greatly appreciate the comments and noted no further edits and suggestions made by reviewer 1.

Sincerely 

RS 

Reviewer 2 Report

In the work, the authors utilized National  Inpatient Sample (NIS) database 2020 to evaluate and compare clinical outcomes in COVID-19 patients with and without PE and discovered that COVID-19 patients with pulmonary embolism had a higher incidence of in-hospital mortality, sudden cardiac arrest, cardiogenic shock, need for vasopressors, mechanical ventilation, and mechanical circulatory support as compared to COVID-19 patients without PE. The overall introduction on background and statistical methods are clearly presented in the manuscript and supporting material. The contents of the manuscript lie within the scope of the Vaccines. However, this manuscript exhibits several weaknesses that should be improved.

1.     COVID-19 patients with PE has worse clinical outcomes than patients without PE, which is  kind of obvious, what the impact of COVID in these clinical outcome is not clearly elucidated, for example  COVID patients with PE VS non-COVID patients with PE, does COVID make it worse or no significant association.

2.     Another interesting topics worth of mentioning is the association of COVID and the incidence of PE and what the triggering mechanism for PE in patients with COVID-19.

3.     Fig.2 shows PE with -ve is higher in mortality which is contradicting with the text

4.     The authors mentioned that “the National Institute 227 of Health recommended the utilization of therapeutic anticoagulation in non-critically ill 228 COVID-19 disease (suggested benefit in favor of organ support-free days and mortality) 229 and prophylactic anticoagulation in severe, critically ill COVID-19 disease [16, 17].”, but without discussing the potential underlying mechanisms and the association with the current study.

5.     Our study found patient with COVID-233 19 and PE are 3 times more likely to go into cardiogenic shock as compare to only COVID-234 19 patients. Again, is this ratio higher, lower or comparable than non-COVID patients with PE. What is the mechanism associate the cardiogenic shock with PE.

Author Response

Dear Reviewer, 

Thank you for your valuable feedback; please see below our point-by-point responses to your queries:

In the work, the authors utilized National  Inpatient Sample (NIS) database 2020 to evaluate and compare clinical outcomes in COVID-19 patients with and without PE and discovered that COVID-19 patients with pulmonary embolism had a higher incidence of in-hospital mortality, sudden cardiac arrest, cardiogenic shock, need for vasopressors, mechanical ventilation, and mechanical circulatory support as compared to COVID-19 patients without PE. The overall introduction on background and statistical methods are clearly presented in the manuscript and supporting material. The contents of the manuscript lie within the scope of the Vaccines. However, this manuscript exhibits several weaknesses that should be improved.

  1. COVID-19 patients with PE has worse clinical outcomes than patients without PE, which is  kind of obvious, what the impact of COVID in these clinical outcome is not clearly elucidated, for example  COVID patients with PE VS non-COVID patients with PE, does COVID make it worse or no significant association.

 Response:

Interesting and thoughtful suggestion by reviewer which could lead to another project with recommended patients’ comparison group. Our study has looked at impact of PE on the outcomes of COVID-19, as we did not study non-COVID PE population as we thought this would not be an ideal comparison group for various reasons. 1. Comparing non- COVID-19 PE, to COVID-19 PE patients, we can argue that COVID-19 infection and PE cohort will have worse mortality due to critical nature of the disease specially in early pandemic (it would be difficult to differentiate whether patients died due to complications of COVID-19 or pulmonary embolism). 2. Patients with PE without COVID-19 can be admitted for any reason like post-surgery and trauma and many other reasons which is not a substitute for COVID-19 infection. 3. Also many patients with PE are discharged from ED on anticoagulation and are not admitted to hospital so we would lose significant sample size.

But as per reviewer suggestion, to describe the impact of COVID on PE outcomes, I have added available literature in discussion by Cossio et al and Miro et al who have already described clinical characteristics and impact of COVID in PE outcomes.

  1. Another interesting topics worth of mentioning is the association of COVID and the incidence of PE and what the triggering mechanism for PE in patients with COVID-19.

 Response:

We have already described the association of COVID-19 and PE in introduction and discussion along with proposed pathophysiology. As per reviewer’s comment, I have further expanded the pathophysiology and association of COVID-19 with PE in the introduction.

  1. 2 shows PE with -ve is higher in mortality which is contradicting with the text

 Response:

Thank you for pointing this out, it was a typo in figure and it has been corrected in the revised manuscript.

  1. The authors mentioned that “the National Institute 227 of Health recommended the utilization of therapeutic anticoagulation in non-critically ill 228 COVID-19 disease (suggested benefit in favor of organ support-free days and mortality) 229 and prophylactic anticoagulation in severe, critically ill COVID-19 disease [16, 17].”, but without discussing the potential underlying mechanisms and the association with the current study.

 Response:

As per reviewer’s suggestion, I have described the mechanism in introduction. And we have elaborated that during our study period anticoagulation use was heterogonous, hence its effects on outcomes cannot be clearly elucidated to this limitation.

  1. Our study found patient with COVID-233 19 and PE are 3 times more likely to go into cardiogenic shock as compare to only COVID-234 19 patients. Again, is this ratio higher, lower or comparable than non-COVID patients with PE. What is the mechanism associate the cardiogenic shock with PE

 Response:

We have already described mechanisms of cardiogenic with COVID related PE but I have elaborated it further. Moreover, we have added cardiogenic shock incidence in non-COVID PE patients and did comparison to our study.

Sincerely, 

RS

Reviewer 3 Report

The manuscript brings a great population to evaluate the importance of pulmonary embolism among hospitalized patients with COVID-19. However, I have several concerns about the study.

1. The study did not analyze vaccines in the entire text. Why did the authors submit the article to Vaccines?

2. All tables should be revised. The authors should cite them as they are cited in the text (e.g. table 2 appears after table 3). In addition, the authors should: a) improve the title of the tables; b) explain the findings presented in the text; c) include the absolute numbers in the tables; d) include all the abbreviations in the legends (in some cases, it is important to reduce the use of abbreviations).

3. Figures. Figure 1. The percentage at the bottom is not equal to the percentages presented in the figure; Figure 2. To increase the quality and to avoid the use of some symbols, such as a tombstone.

Minor comments

To revise the overall presentation (e.g. Covid 19 - COVID-19; 46440 - 46,440)

To include the * in the corresponding author-name

To correct ≥18-29 to ≤18-29

Importantly, in my opinion, the manuscript should be corrected and submitted to other MDPi journals.

Author Response

Dear Reviewer, 

Thank you for your suggestions and feedback. Please see below our point-by-point reply to your queries

The manuscript brings a great population to evaluate the importance of pulmonary embolism among hospitalized patients with COVID-19. However, I have several concerns about the study.

  1. The study did not analyze vaccines in the entire text. Why did the authors submit the article to Vaccines?

 Response:

Our study analyzed patients before the administration of COVID-19 vaccines. Reviewer has a great suggestion, effect of vaccination on COVID-19 patients with PE can be assessed in future studies. As per reviewer’s comments, I have added available literature on efficacy of vaccines in prevention of COVID associated PE. Also, we reached out to Editor in Chief for the special issue: COVID-19 Epidemiology and Transmission for suitability of our manuscript and after their approval we submitted this manuscript. Similar sort of studies looking at COVID-19 infection outcomes have been published previously in this special issue and the journal of Vaccines.

  1. All tables should be revised. The authors should cite them as they are cited in the text (e.g. table 2 appears after table 3). In addition, the authors should: a) improve the title of the tables; b) explain the findings presented in the text; c) include the absolute numbers in the tables; d) include all the abbreviations in the legends (in some cases, it is important to reduce the use of abbreviations).

 Response:

We have made edits as suggested by the reviewer except adding absolute numbers in the table as it will make table quite busy and difficult to comprehend as a group our goal is to make the tables reader friendly, great suggestion we initially did added absolute numbers but it appeared very crowded so we took it off. Thank you!

  1. Figures. Figure 1. The percentage at the bottom is not equal to the percentages presented in the figure; Figure 2. To increase the quality and to avoid the use of some symbols, such as a tombstone.

 Response:

We have made edits as suggested by the reviewer. It is not cumulative percentage but various comorbidities in each subset hence we removed the 100% at bottom of the figure Thank you!

Minor comments

To revise the overall presentation (e.g. Covid 19 - COVID-19; 46440 - 46,440)

To include the * in the corresponding author-name

To correct ≥18-29 to ≤18-29

 Response:

We have made edits as suggested by the reviewer. Thank you!

Sincerely,

RS

Round 2

Reviewer 3 Report

Title: COVID-19 and Pulmonary Embolism Outcomes Among Hospitalized Patients in the United States: A Propensity Matched Analysis of National Inpatient Sample

Abstract

(i) to change “COVID-19 patients” to “Patients with COVID-19”

(ii) to exclude the abbreviation from the Keywords

Introduction
(i) to edit the text ad references (e.g. “with COVID-19” should be “with COVID-19.” / “non-COVID patients” should be “non-COVID patients” / “male gender” should be “
male sex”

Methods

(i) Important: The authors should include the data from 2021 and 2022

(ii) To change “COVID-19 infection” to “SARS-CoV-2 infection”

(iii) To include a correction by multiple tests.

Results

(i) To change “COVID-19 patients” to “Patients with COVID-19”

(ii) To exclude the words “pulmonary embolism”

(iii) Table 1. To include the abbreviations in the legend. It is necessary to include absolute numbers. To change “Covid 19” to “COVID-19”. To edit the layout because it is difficult to interpret the table.

(iv) To revise the abbreviations in the text.

(v) Figure 1.To improve the resolution. In addition, the authors can use the same groups as table 1 (COVID-19 without pulmonary embolism versus COVID-19 with pulmonary embolism)

(vi) Tables 2-5. To include the same corrections as Table 1.

(vii) Figure 2. It was not possible to see all information.

Author Response

Dear reviewer please see our comments for your queries. 

Abstract

  • to change “COVID-19 patients” to “Patients with COVID-19.”

Response to the reviewer- Thank you for bringing this to our attention. This has been modified.

(ii) to exclude the abbreviation from the Keywords

 Response to the reviewer- Thank you for bringing this to our attention. This has been modified.

Introduction
(i) to edit the text ad references (e.g. “with COVID-19” should be “with COVID-19.” / “non-COVID patients” should be “non-COVID patients” / “male gender” should be “male sex”

Response to the reviewer- Thank you for bringing this to our attention. This has been modified. 

Methods

  • Important: The authors should include the data from 2021 and 2022

Response to the reviewer- Dear reviewer, we acknowledge the study's limitation and have included data only from 2020. However, given the relative delay in the availability of NIS data, only data from 2020 is currently available. This data was recently released in October 2022. We hope as the data from 2021 and 2022 become available, future studies can build upon our findings.

  • To change “COVID-19 infection” to “SARS-CoV-2 infection”

Response to the reviewer- Dear reviewer, thank you for your suggestion. We have included a statement clarifying that COVID-19 infection is caused by SARS-CoV-2 virus. However, if feasible, we would like to keep it as COVID-19 infection, as it is the more accepted and widely known terminology used in the majority of published literature, including by CDC.

(iii) To include a correction by multiple tests.

Response to the reviewer- we conducted a secondary analysis with propensity matching to confirm results obtained by traditional multivariate analysis. PSM results for COVID-19 positive patients with and without PE were similar to numbers in regards to in-hospital mortality and complication  

Results

  • To change “COVID-19 patients” to “Patients with COVID-19.”

Response to the reviewer- Thank you for bringing this to our attention. This has been modified.

  • To exclude the words “pulmonary embolism”

Response to the reviewer- Thank you for bringing this to our attention. This has been modified.

  • Table 1. To include the abbreviations in the legend. It is necessary to include absolute numbers. To change “Covid 19” to “COVID-19”. To edit the layout because it is difficult to interpret the table.

Response to the reviewer- Thank you for bringing this to our attention. This has been modified

  • To revise the abbreviations in the text.

Response to the reviewer- Thank you for bringing this to our attention. This has been modified.

  • Figure 1.To improve the resolution. In addition, the authors can use the same groups as table 1 (COVID-19 without pulmonary embolism versus COVID-19 with pulmonary embolism)

Response to the reviewer- Thank you for bringing this to our attention. This has been modified.

  • Tables 2-5. To include the same corrections as Table 1.

Response to the reviewer- Thank you for bringing this to our attention. This has been modified.

  • Figure 2. It was not possible to see all information.

Response to the reviewer- Thank you for bringing this to our attention. This has been modified.

Kind Regards 

RS 

Round 3

Reviewer 3 Report

I still believe that the manuscript is outside the scope of the journal. However, the authors answered the questions, and if the editor deems it appropriate, the manuscript can be published.